# Behavior Tracking and Analyses of Group-Housed Pigs Based on Improved ByteTrack

**DOI:** 10.3390/ani14223299

**Published:** 2024-11-16

**Authors:** Shuqin Tu, Haoxuan Ou, Liang Mao, Jiaying Du, Yuefei Cao, Weidian Chen

**Affiliations:** 1College of Mathematics and Informatics, South China Agricultural University, Guangzhou 510642, China; tsq5_6@scau.edu.cn (S.T.); ouhaoxuan123@163.com (H.O.); dujiaying2001@163.com (J.D.); cyf_1208@163.com (Y.C.); chenwd2543@163.com (W.C.); 2Institute of Applied Artificial Intelligence of the Guangdong-Hong Kong-Macao Greater Bay Area, Shenzhen Polytechnic University, Shenzhen 518055, China

**Keywords:** behavioral analysis algorithm, multi-object tracking (MOT), long-term video tracking, Pig-ByteTrack

## Abstract

This study develops a new method to automatically track and analyze pig behavior in complex environments. We use Pig-ByteTrack algorithm for real-time tracking and perform trajectory interpolation post-processing on the original algorithm to solve tracking problems caused by light changes, occlusion, and collisions between pigs. Finally, a set of time statistical algorithms for behavior categories are designed. The method helps farm managers detect abnormalities and health problems of pigs in a timely manner. Experimental results show that the method performs well in behavior recognition and tracking, accurately records pig behavior, and provides technical support for monitoring the health and welfare of pig herds.

## 1. Introduction

With the increasing demand for animal products and the growing social concern for animal welfare, effective monitoring and analysis of animal welfare is increasingly becoming a hot research priority. The health status of pigs will determine the development and economic efficiency of pig farming. Meanwhile, clinical or subclinical signs of most swine diseases are usually accompanied by behavioral abnormalities before the appearance of symptoms [1]. Currently, with the development of image processing technology, the integration of manual observation and computer vision monitoring is the main management method in large-scale pig farms, this still requires a large amount of labor. And with the rapid development of Multi-Object Tracking (MOT) technology in the field of video surveillance, the need for manual labor can be significantly reduced through the application of MOT for the pig industry, improving the efficiency and cost-effectiveness of pig farm management.

In recent years, several outstanding MOT algorithms have been proposed. For example, the prevailing trackers including simple online and real-time tracking (SORT [2]) and deep simple online and real-time tracking (DeepSORT [3]), have been widely used for the MOT of pedestrians and vehicles. SORT is a data association method based on a Kalman filter (KF) and a Hungarian algorithm to associate the detected bounding-box results between adjacent frames. DeepSORT introduces the comparison of appearance features, which is added to the motion model in SORT. This enhances the performance for a longer duration of occlusion. Zhang et al. proposed Fair Multi-Object Tracking (FairMOT) based on the anchor-independent target detection architecture CenterNet, which obtains high accuracy in detection and tracking for MOT datasets [4]. Sun et al. proposed an end-to-end Transtrack MOT method based on TransFormer, which can perform target detection and tracking tasks simultaneously [5]. The framework performed well in complex scenarios and achieved excellent performance on MOT16, MOT17, and MOT20 datasets. Zhang et al. proposed the ByteTrack algorithm, which adds a correlation phase to low-scoring detection frames to improve tracking performance on many pedestrian tracking datasets [6]. Aharon et al. proposed the Bayesian online tracking with sorting (BOT-SORT) algorithm, which achieves accurate tracking of multiple targets by using appearance and motion information for re-identification [7]. It could effectively deal with challenges such as target occlusion and appearance changes. Due to the excellent performance of these MOT algorithms, they are widely applied in various fields, including autonomous driving, traffic management, drones, aviation, medical image analysis, agriculture [8,9,10,11,12,13,14,15,16,17], and so on.

Nowadays, more and more studies of MOT technology have been used in livestock detection and tracking. For example, Guo et al. improved the joint detection and embedding (JDE), FairMOT, and you only look once version 5 small (YOLOv5s) + DeepSORT algorithms, respectively, to improve the performance of individual animal tracking, especially to reduce the number of identity switches, thus ensuring timely animal welfare [18]. Zheng et al. proposed a MOT method (YOLO-BYTE), which aims to address the problem of missed and false detections caused by the complex environment in the detection and tracking of individual cows [19]. Lu et al. proposed a MOT method based on a rotating bounding box detector to recognize the number of pigs and monitor their health status, which improved the adaptability of the tracking technique in complex scenarios and reduced the switching of pig identities [20]. Zou et al. proposed an improved YOLOv3+DeepSORT algorithm to achieve accurate MOT of yellow-feathered broilers in large-scale broiler farms, which provides a technical reference for analyzing the behavioral perception and health status of broilers [21]. However, all these methods focused on target detection and tracking tasks, little research was conducted for further behavioral automated analysis based on the MOT results.

In addition, most work regarding pig detection and tracking involves studies on the pig tracking of short-duration videos or identifying postural behaviors. For example, Alameer et al. utilized a MOT technique for automated diagnosis and intelligent monitoring of health status in 1 min videos of pigs on a pig farm [22]. Huang et al. designed the HE-Yolo (High Effect Yolo) model to identify the postural behaviors of fenced pigs in real-time from 10 to 100 s of video [23]. Huang et al. proposed an improved pig counting algorithm (MPC-YD) based on theYOLOv5 + DeepSORT model, which aims to solve the problems of partial feature detection, tracking loss due to fast movement, and video counting errors of pigs [24]. Our previously published work proposed an improved DeepSORT algorithm for multi-target behavioral tracking in 1 min pig videos. The approach could improve tracking accuracy under complex scenarios and reduce error IDs due to overlapping and occlusion between pigs [25]. All the above methods demonstrated superior performance in pigs’ MOT tasks for short-time videos. However, there is little research on pigs’ behavioral tracking for long-time videos. This is because in the long-term MOT of pigs, factors such as the occlusion of objects and influence of light appear with greater probability, which leads to a decrease in the accuracy and stability of detection and tracking of pigs; it is very challenging to recognize the behavior of pigs on this basis. Therefore, it is extremely challenging to record the behaviors of individual pigs for the long term completely by MOT methods.

To address the above challenges, we proposed an approach to complete target detection, tracking, and behavioral automated analysis of three tasks for group-housed pigs. First, we used the YOLOX-X detector in the detection task to detect the target and output for the four behavioral categories (lie, stand, eat, and other), locations, and confidence values of the pigs. Then, we employed a trajectory interpolation post-processing strategy in the data association part to minimize ID errors caused by occlusion to improve the tracking accuracy. Finally, we designed an automated behavioral analysis algorithm to calculate four behavioral times of each pig in each pen based on its ID and behavioral information.

In this study, our main objective was to develop an automated method for monitoring and analyzing the behavior of group-reared pigs to detect health problems and improve animal welfare promptly. Specific objectives include the following:(1)We proposed the Pig-ByteTrack algorithm, integrating trajectory interpolation to reduce false alarms and enhance tracking stability, to improve the accuracy of behavior monitoring in pig farming.(2)We designed a behavioral analysis algorithm to calculate the temporal distribution of behaviors for each pig, leveraging tracking IDs and categories, to enable detailed behavioral analysis within individual enclosures.(3)We constructed a 10 min long-term pig dataset with real pig house videos and validated our methodology’s effectiveness through comparative tracking experiments, to ensure the practical applicability and reliability of our approach under real-world conditions.

## 2. Method

To address the challenges of accurate precision in pig tracking and behavioral analyses under complex environments, we proposed the process of tracking and behavioral time statistics for group-housed pigs, as depicted in Figure 1. Firstly, we introduced a novel target tracking algorithm called Pig-ByteTrack, which is used to detect the pigs, classify the behavior categories, and assign the target pig ID for input video sequences. Then, the MOT results were generated, which included the following three parts: pig ID, the pigs’ location, and the pigs’ behavioral categories. Additionally, a behavioral analysis algorithm was designed to calculate the frequency of each behavior for each pig based on the pig’s ID and behavior category information. Finally, based on a behavioral analysis algorithm, we designed the program to visualize the frame number, categories, and frequency for each pigs’ behaviors, thereby obtaining statistical results of pig behaviors in the videos.

### 2.1. Pig-ByteTrack Algorithm of Group-Housed Pigs

Pig-ByteTrack is divided into two main phases: object detection and multi-target tracking. The workflow is shown in Figure 2. Firstly, the YOLOX-X detector is used to detect all pig targets for each frame of input video sequences, and the detection results contained pigs’ confidence value, bounding box (BB), and behavioral category. Then, the MOT tracker uses the improved BYTE data association algorithm to match the high-scoring detection boxes to the trajectory, and the unmatched trajectory is linked to the low-scoring boxes. The improved Byte includes two operations: the Hungarian Matching Algorithm and the Kalman Filter (KF) prediction, respectively. Finally, the target trajectory output for the images of consecutive video frames is obtained by the MOT tracker.

#### 2.1.1. The Original ByteTrack Algorithm

The ByteTrack algorithm utilized the detector of YOLOX to complete the BB regression and behavior recognition. The detector of YOLOX is an improved version of YOLO [26,27] with a simple scheme and better performance without the anchor mechanism. YOLOX decoupled the YOLO detection head into distinct feature channels for box coordinate regression and target classification. Then, the ByteTrack algorithm utilized the BYTE data association strategy for MOT task. Its processing flow chart is shown in Figure 3.

The detailed process is as follows:(1)The object detection results are divided into high-score and low-score detection boxes.

In the detection results, if the confidence value of the detection box is greater than the high score box threshold, the detection box is placed in the high score detection box set ($D_{high}$). If the confidence value of the detection box is less than the high score box threshold and greater than the low score box threshold, the detection box will be placed in the low score detection box set ($D_{low}$).

(2)The high-scoring detection boxes $D_{high}$ are matched with the existing tracks for the first IoU data association.

The IoU distance matrixes between high-scoring detection boxes $D_{high}$ and the set of trajectories are calculated and then used to match using the Hungarian algorithm, which produces three kinds of outputs including matched track set, unmatched high-scoring detection boxes, and unmatched tracks. The matched track set contains successfully matched detection boxes updated with their Kalman filter. The unmatched high-scoring detection boxes and unmatched tracks are placed in $D_{remain}$ and $T_{remain}$ sets, respectively.

(3)The low score detection boxes $D_{low}$ are associated with the unmatched trace-in $T_{low}$ for the second IoU data.

We calculate the IoU distance matrixes between $D_{low}$ and the unmatched tracks $T_{remain}$, which output three kinds of sets, including unmatched tracks, unmatched low-scoring detection boxes, and matched track sets. The unmatched trajectories are placed in $T_{lost}$ set. The unmatched low-score detection boxes will be considered background boxes and will be deleted directly. For the set of successfully matched trajectories, its Kalman filter is updated and placed in the current frame trajectory set.

(4)Trajectories creation, deletion, and merging.

For the detection boxes $D_{remain}$, if the confidence values are greater than the tracking threshold, new tracks are created for them and are merged into the current set of trajectories for the current frame, if not, they are neglected. The stored trajectories that exceed 30 frames in $T_{lost}$ are deleted. All trajectories for the current frame are outputted and fed to the next image frame as the existing trajectories.

#### 2.1.2. The Pig-ByteTrack Algorithm

To implement stable behavior tracking of group-housed pigs, based on ByteTrack, the improvement of the Pig-ByteTrack algorithm consists of two steps as follows:(1)The design of suitable detection anchor boxes and the improvement of tracking boxes.

The anchor boxes of the original detection of the ByteTrack tracker are designed based on the pedestrian’s features of narrow height which is not suitable for group-housed pigs. We remove the shape restriction on the pig detection box and implement the most appropriate ratio of anchor boxes for pigs. At the same time, four behavioral classes of pigs (lie, stand, eat, and other) are added to the BYTE tracker for tracking.

The visualization of the Pig-ByteTrack and ByteTrack is shown in Figure 4. The tracking box of the original ByteTrack in Figure 4a only shows the ID number of each pig, whereas the tracking box of the Pig-ByteTrack in Figure 4b can reflect the behavioral categories and ID value of each pig at the same time.

(2)The implementation of the trajectory interpolation post-processing strategy for BYTE tracker.

To avoid error IDs due to severe occlusion between group-housed pigs, we propose the trajectory interpolation post-processing strategy to significantly improve the stable tracking performance of occluded targets. The process is as follows:

Suppose a trajectory T is lost due to occlusion between $t_{1}$ frame and $t_{2}$ frame, if the current trajectory T is at exactly t frame ($t_{1}<t<t_{2}$), the interpolation box $B_{t}$ of the trajectory T can be calculated by Equation (1) as follows:(1)$$B_{t}=B_{t1}+(B_{t2}-B_{t1})\frac{t-t_{1}}{t_{2}-t_{1}}$$
where $B_{t}$ denotes the track box coordinates of the t frame (containing four values, one for the top left and one for the bottom right coordinates). Moreover, $B_{t1}$ denotes the track box coordinates of the $t_{1}$ frame, $B_{t2}$ denotes the track box coordinates of the $t_{2}$ frame.

### 2.2. The Behavioral Analysis Algorithm

Based on the video sequence tracking results, we design and implement the pig’s four behavioral time calculation algorithm as shown in Algorithm 1. The algorithm flow is as follows:(1)An array of a behavioral category named $\left[A_{1},A_{2},A_{3},A_{4}\right]$ is designed for each track, which creates statistics for the number of all tracks for the four categories of the stand, lie, eat, and other behaviors. The statistic is added as an attribute in each of the tracks.(2)For each frame of video, the BYTE data association algorithm first obtains information about the YOLOX-X detection results of each BB named D, including the category information mentioned above. Then, the behaviors analysis algorithm creates an array of frames named $\left[a_{1},a_{2},a_{3},a_{4}\right]$ based on the categories (stand, lie, eat, and other behaviors of each BB) for each pig ID. And if the behavior category belongs to stand, *a*_1_ is set to 1 and the other parameters are set to 0, and so on.(3)After Associating T with D using the BYTE operation, we can revise the values $\left[A_{1},A_{2},A_{3},A_{4}\right]$ if the tracklet and detection BB match successfully. The formula of revised A is as follows:(2)$$\left[\begin{array}{c} A_{1new}\\ A_{2new}\\ A_{3new}\\ A_{4new} \end{array}\right]=\left[\begin{array}{c} A_{1}\\ A_{2}\\ A_{3}\\ A_{4} \end{array}\right]+\left[\begin{array}{c} a_{1}\\ a_{2}\\ a_{3}\\ a_{4} \end{array}\right]$$

If there is no match between the detection BB and the track, or if the confidence value of the detection BB is greater than the high score threshold, we can set the value of $\left[A_{1},A_{2},A_{3},A_{4}\right]$ for this tracklet to 0. Finally, after summing the number of frames obtained for the four behaviors with different pig IDs, we divide it by the frame rate to obtain the time for the different behaviors of each pig ID.
**Algorithm 1.** Pseudo-code of Behavior Category Time Statistics of Pigs**Input:** A video sequence V; object detector Det; the k frame $f_{k}$; the detection BB D; category includes lie, eat, stand, other; variable a and A: one-dimensional array including four elements for time statistics; T: tracklet information and four behavior category time statistics; tracking score threshold η is set 0.75; Frames per second Fps;**Output:** Tracks T of the video Initialization: T←∅**for** frame $f_{k}$ in *V* **do**   $D\leftarrow Det(f_{k})$  Initialize time-count array including four elements for time statistics a←[0,0,0,0]  Set variable category_index←D{category}  [category_index]←1  Associate T with D using BYTE:    **if** succeed to match **then**      Call the Update or Re-activate function to update the status of tracks      Set variable $A\leftarrow T\left\{category\_\_time\_array\right\}+a$      $T\left\{category\_\_time\_array\right\}\leftarrow A$    **end**    **if** *D* failed to match and D> **then**      Call the function to create a new track      Initialize time-count array A←[0,0,0,0]       $T\left\{category\_\_time\_array\right\}\leftarrow A$    **end**end $T\left\{category\_\_time\_array\right\}=T\left\{category\_\_time\_array\right\}/Fps$Return *T*

## 3. Experiment

All the experiments in this study were conducted on the same computer using Linux as the experimental platform with Ubuntu 20.04 operating system, using Python 3.7 as the programming language, Pytorch 1.9.1 as the deep learning framework, and CUDA version 11.1. The GPU server is RTX3090, and the memory is 64 GB. We select HOTA, MOTA, IDF1, and IDs as the pig MOT evaluation metrics.

HOTA calculation is shown in Equation (3) as follows:(3)$$HOTA=\sqrt{\frac{\underset{c\in TP}{\Sigma }A\left(c\right)}{TP+FN+FP}}$$
where c is a point belonging to *TP*, according to which we can always determine a unique Ground Truth trajectory, and $A\left(c\right)$ represents the association accuracy. TP refers to the number of positive samples; FN refers to the positive samples predicted by the model to be negative; and FP represents the negative samples predicted by the model as positive samples.

MOTA calculation is shown in Equation (4) as follows:(4)$$MOTA=1-\frac{\sum_{t}\left(FP+FN+IDS\right)}{\sum_{t}g_{t}}$$
where FP represents the total number of false detections in frame *t*; FN represents the total number of missed detections in frame *t*; IDS represents the number of times the target label ID switched during tracking in frame *t*; and *g_t_* represents the number of targets observed at frame *t*.

IDF1 calculation is shown in Equation (5) as follows:(5)$$IDF1=\frac{2IDTP}{2IDTP+IDFP+IDFN}$$
where IDTP represents the total number of targets correctly tracked with unchanged *ID*; IDFP represents the total number of targets incorrectly tracked with unchanged *ID*; and IDFN represents the total number of targets lost in tracking with unchanged *ID*.

Additionally, the model performance is evaluated with the number of *ID* Switches (*IDs*) in this study. Higher values of HOTA, MOTA, and IDF1, and a lower value of *IDs* indicate better model performance.

### 3.1. Dataset

For a comprehensive analysis of group-housed pig behavior, this study collected two sets of pig behavior video datasets from different scenarios, categorized as public and private datasets. The public dataset [28] comprised video clips of pigs of different breeds, recorded in both daytime and nighttime environments, with each pigpen house including 7 to 20 pigs, with a total of 4 annotated 10 min videos. The private dataset was collected in September 2022 from a commercial pig farm in Foshan, where each pigpen houses 6 to 11 black and spotted pigs. Pigs in the study were categorized into three age groups: nursery (3–10 weeks), early fattening (11–18 weeks), and late fattening (19–26 weeks). Weight distributions across these stages align with standard growth curves. In the nursery phase, pigs weighed 7–10 kg at 3 weeks, 10–15 kg at 4–5 weeks, and 15–25 kg at 6–10 weeks. Early fattening pigs weighed 25–35 kg at 11 weeks, 35–50 kg at 12–15 weeks, and 50–70 kg at 16–18 weeks. In the late fattening phase, pigs weighed 70–90 kg at 19 weeks, 90–110 kg at 20–23 weeks, and 110–130 kg at 24–26 weeks. These estimates may vary due to breed, husbandry practices, and diet. The study included two breeds: black and spotted pigs. Welfare principles were adhered to, with each pig allocated 1.2 square meters of pen space. The housing environment was controlled for temperature and humidity to ensure suitability for the animals, and the facility was equipped with adequate ventilation for herd density. Bedding conditions comprised both solid and metal grid flooring. Empirical evidence suggests that pigs display comparable behaviors on solid and grid floors, indicating that flooring preferences do not significantly affect basic behavioral patterns and thus do not impact the study’s findings.

This dataset annotated 18 1 min video segments, with 9 video segments for training and 9 for testing. Each video segment had a frame rate of 5 frames per second, resulting in 300 images in a 1 min video and 3000 images in a 10 min video segment. All datasets were annotated using DarkLabel1.3 software. Our classifications of pig activity levels were determined by direct observation in cooperation with farm staff. Specifically, the classifications of ‘low’, ‘medium’, and ‘high’ activity levels were based on daily observations and empirical judgments of staff on pig behavior. These categories, although subjective, reflect the routine practice of actual pig farming and provide a practical benchmark for our study. In future studies, we plan to introduce more objective quantitative methods to enhance the accuracy and consistency of the categories. We divided the activity level of pigs into three categories according to the time period: high activity during the day, medium activity during the day (or night), and low activity during the day (or night). The activity level of pigs was defined as follows: during the day (07:00–17:00), pigs have more frequent behaviors such as eating and walking, this time was defined as a high activity level. During the day (07:00–17:00) or night (18:00–06:00), pigs have no high activities such as eating and walking. This time was defined as the medium activity level during the day or night. During the day (07:00–17:00) or night (18:00–06:00), pigs had few eating and walking behaviors, mostly lying behavior. This time period was defined as low activity during the day or night. The detailed test dataset is shown in Table 1.

The experiment consisted of the following four parts: (1) the comparison of tracking results of Pig-ByteTrack, ByteTrack, and TransTrack in the private dataset; (2) the analysis of MOT results using Pig-ByteTrack for 1 min videos in the private dataset; (3) the tracking results analysis of Pig-ByteTrack for 10 min videos; and (4) the behavioral analysis for four video segments in the private dataset.

### 3.2. Experimental Results and Performance Comparison

In this section, we evaluated the achievement of experiment goals in detail. To achieve these goals, the following research tools and methods were used:(1)The YOLOX-X detection model was used for target detection and behavioral recognition in pigs.(2)The Pig-ByteTrack algorithm was designed to track behavioral information for each pig.(3)The Automated Behavioral Analysis algorithm was developed to calculate the temporal distribution of behaviors.

#### 3.2.1. Results Comparison of Pig-ByteTrack, ByteTrack, and TransTrack

The results comparison of the Pig-ByteTrack with the ByteTrack and TransTrack in the private dataset is shown in Table 2. The Pig-ByteTrack achieved the best performance with HOTA, MOTA, IDF1, and IDs of 72.9%, 91.7%, 89% and 41, respectively. Compared with the ByteTrack, the results of the Pig-ByteTrack improved by 1.5%, 1.1%, 1.1% and 14 in HOTA, MOTA, IDF1, and IDs, respectively. Compared with TransTrack, its HOTA, MOTA, and IDF1 were improved by 23.4%, 4.4%, and 21%, and its IDs decreased 212, respectively, with significant performance improvement. Combining the above results, we found that Pig-ByteTrack can achieve stable behavioral tracking of group-reared pigs.

The tracking results for Pig-ByteTrack, ByteTrack, and TransTrack in Videos 10, 11, 12, and 13 were shown in Figure 5. The Pig-ByteTrack method not only provided the ID number but also the behavioral class of each tracked pig, whereas ByteTrack and TransTrack only displayed the pig’s ID without behavioral categorization. Pig-ByteTrack consistently achieved accurate pig behavior tracking across all four videos, with minimal IDs. In Video 10, ByteTrack failed to accurately identify a pig within the red dashed box at frame 17. In Video 11, ByteTrack incorrectly changed an ID indicated by the arrow in frame 26, where the pig was changed from ID11 to ID12. TransTrack missed detecting two black pigs within the red dashed box. Video 12 was captured in a challenging night scene, where the color of the pig is very similar to the shadow, making it difficult to identify the pig. Pig-ByteTrack had no missed detection, and ByteTrack and TransTrack both exhibited missed detection within the red dashed box. In Video 13, the same problem occurred with ByteTrack and TransTrack.

#### 3.2.2. Results of Pig-ByteTrack for 1 min Video in Private Dataset

Pig-ByteTrack performed well in the various video tests with high averages. Table 3 shows its performance in each test video. Pig-ByteTrack performed well in Videos 14, 15, 16, and 17 with HOTA scores of 77.1%, 81.7%, 82.5%, and 79.6%, respectively, and MOTA scores consistently above 97%. In Video 17, both HOTA and IDF1 obtained max values of 97.9% and 99.0%, respectively, with zero IDs. These results showed Pig-ByteTrack could achieve good accuracy in this scenario. However, in Video 18, Pig-ByteTrack performed relatively poorly with 56.0%, 67.7%, and 64.2% for HOTA, MOTA, and IDF1, respectively, and the number of IDs was as high as 17. These results suggested that Pig-ByteTrack is affected in scenes with low lighting at night. Overall, Pig-ByteTrack showed excellent accuracy and stability in the pig MOT task. Further optimization and improvement is needed in complex situations such as severe occlusion and insufficient light.

Figure 6 shows the tracking results for Pig-ByteTrack, ByteTrack, and TransTrack from frames 6 to 116 of Video 11, where the real number of pigs was 11. The Pig-ByteTrack method achieved good tracking throughout and additionally provided behavioral categories for each pig tracked. ByteTrack and TransTrack both experienced tracking loss (pigs in the red dashed box in Figure 6). The tracking performance of Pig-ByteTrack was more stable and accurate than that of the other two methods.

#### 3.2.3. Results of Pig-ByteTrack for 10 min Video Dataset

The results of Pig-ByteTrack in the 10 min videos are shown in Table 4. We found that the tracking results of the 10 min videos were much less than those of the 1 min videos by comparing the results with Table 3. Video 02 had the highest HOTA rate, reaching 69.1% precision, while Video 03 had an HOTA rate of only 50.8%. The reason was that the pig’s active level in Video 03 was the highest among the four videos, as the pig moves frequently, resulting in frequent problems such as target occlusion, motion blurring, or lighting changes. The Pig-ByteTrack achieved the best IDF1 value with 67.0% in Video 04 and the worst IDF1 value with 47.8% in Video 01. The reason for this phenomenon is that Video 04 is in the nighttime period, and most of the pigs are resting.

Figure 7 shows the visualized tracking results of Pig-ByteTrack for a 10 min video. The 10 min video had many issues with lost tracking and IDs (as indicated by the red arrow in Figure 7). We thought that this was because the 10 min video produces the situations such as many pig occlusions, light changes, and so on. The probability of its problems in the 10 min videos is greater than that of the 1 min videos, which leads to the gap in tracking results. Therefore, the current method is still difficult to meet the demand of tracking objects for a long time, which is a direction that needs to be improved in the future.

#### 3.2.4. Behavioral Analysis for Four Video Segments in Private Dataset Algorithm

The histogram results using our designed behavioral analysis algorithm are shown in Figure 7. It presented the number of frames and distribution of different behavior classes indicated by each ID number results for the 1 min time videos, including Videos 14, 15, 16, and 17. In each sub-graph of Figure 7, the horizontal coordinates represented the ID number of the pigs, and the vertical coordinates represented the number of frames for each behavior. Four different colors represented the four behavioral classes of pigs: stand, lie, eat, and other, which were represented by light blue, dark blue, red, and yellow, respectively.

When we used the Pig-ByteTrack method to count the frame numbers of different behaviors of group-housed pigs, the loss of IDs was encountered, but this did not affect the consistency of the statistical results with the actual situation. In some time periods, pigs were masked, resulting in changes in identity, such as id3 and id7 in video15; and id6 in video16. In video14 of Figure 8, id1~4 mainly performed eat behaviors, while id5 and id6 performed lie and other behaviors, respectively, which proved that the pigs were highly active in this video. Moreover, in video15 of Figure 8, the pigs mainly engaged in eat and lie behaviors, with a similar proportion of the two, which indicated that the pigs were generally active. For video16 in Figure 8, most of the pigs exhibited stand behaviors, and only id3 and id5 engaged in other behaviors, which inferred that the pigs were medium active. Lastly, in video17 in Figure 8, the pigs, except id5, mainly exhibited stand behavior, proving the low activity level of the pigs. According to Table 1, we found that the results of videos14~17 are completely in line with the actual situation.

By the pig behavior statistics algorithm, our research provides professionals with a tool that enables them to identify and initially determine the cause of abnormal behavior. For example, if our system monitors a pig not eating for an extended period of time, this may indicate that the pig is experiencing a loss of appetite. This change in behavior could be due to a health issue, environmental discomfort, or a feed problem. If a pig is monitored lying down for an extended period of time without normal activity, this may indicate a possible injury or discomfort. With this behavioral data, our system helps farm managers and veterinary professionals quickly identify pigs of concern and provide an initial indication of the cause of abnormal behavior. This provides valuable time for further diagnosis and timely intervention.

## 4. Conclusions

In this paper, a Pig-ByteTrack method was proposed for target detection, behavioral classification, and multi-target tracking of group-reared pigs. The Pig-ByteTrack method achieves an HOTA of 72.9%, an MOTA of 91.7%, an IDF1 of 89%, and IDs of 41 for the behavioral tracking of pigs. Compared with ByteTrack, the HOTA, MOTA, IDF1, and IDs of the Pig-ByteTrack method were improved by 1.5%, 1.1%, 1.1%, and 14, respectively. The advantage of the Pig-ByteTrack method was tracking all detection frames and then dividing the detection frames into high-scoring and low-scoring detection frames. Meanwhile, the post-processing strategy of trajectory interpolation is used in data correlation to maximize the accuracy of tracking under occlusion. Based on Pig-ByteTrack, we designed an algorithm for statistical analysis of pig behavior. The algorithm could calculate the number of times each pig behaves in each pen according to the pig’s ID and category information and could visualize the behavioral statistics time of pigs. Finally, we conducted a multi-target tracking study on the 10 min-long video of a pig, which provided a technical reference for subsequent long-time video tracking studies in this field.

Through the analysis of the method in this paper, we could consider that future research could focus on enhancing the model’s long-term tracking performance. This could be achieved by delving deeper into the optimization of deep learning models, which entails refined network structures and fine-tuning hyperparameters. Furthermore, the integration of advanced data analysis and machine learning techniques presented an opportunity to develop predictive models for assessing the health status of pigs based on their behaviors. By continuously monitoring various parameters such as activity levels, feeding patterns, movement trajectories, and physiological indicators like body temperature and heart rate, it became feasible to implement real-time tracking systems for monitoring the health conditions of the pigs.

## Figures and Tables

**Figure 1 animals-14-03299-f001:**
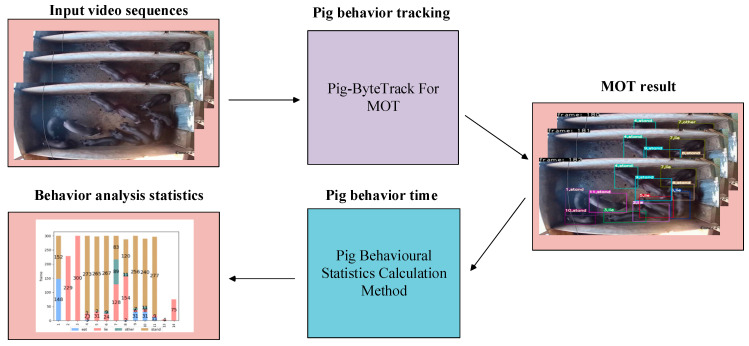
Process diagram of tracking and behavioral time statistics for group-housed pigs.

**Figure 2 animals-14-03299-f002:**
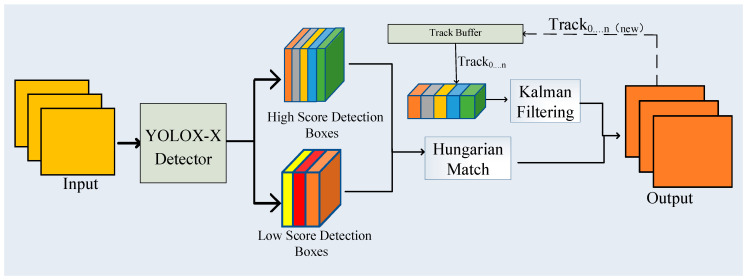
Flow chart of Pig-ByteTrack algorithm.

**Figure 3 animals-14-03299-f003:**
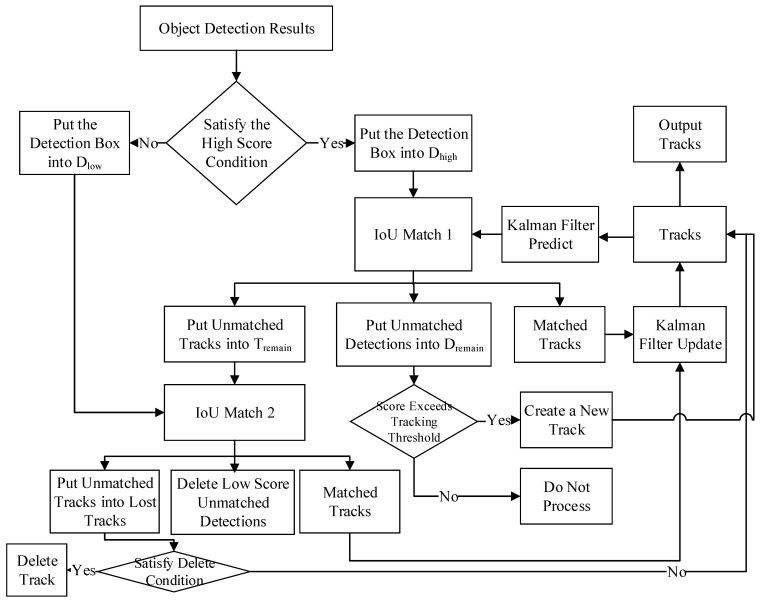
The flow chart of the Byte data association algorithm.

**Figure 4 animals-14-03299-f004:**
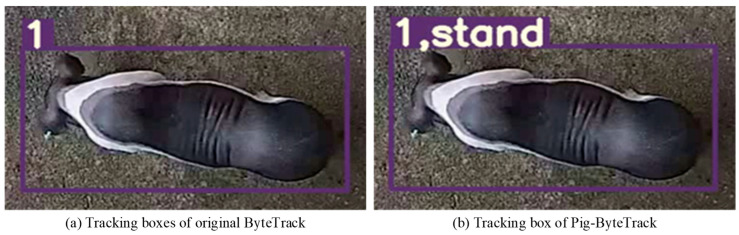
Comparison of tracking box between Pig-ByteTrack and ByteTrack.

**Figure 5 animals-14-03299-f005:**
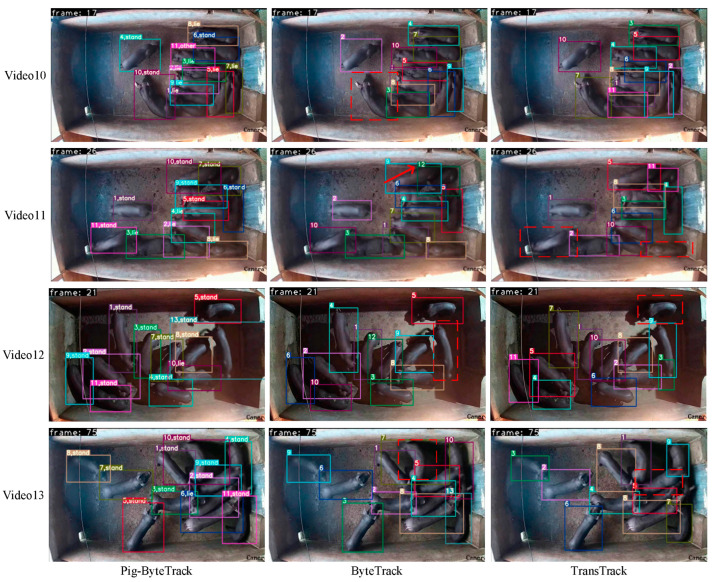
Comparison of Pig-BytetTrack, ByteTrack and TransTrack results on private datasets.

**Figure 6 animals-14-03299-f006:**
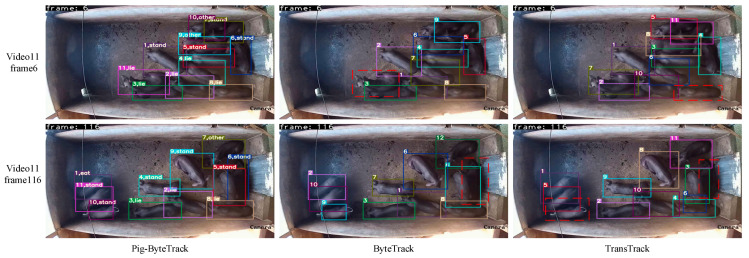
The visualized tracking results comparison of Pig-BytetTrack, ByteTrack, and TransTrack.

**Figure 7 animals-14-03299-f007:**
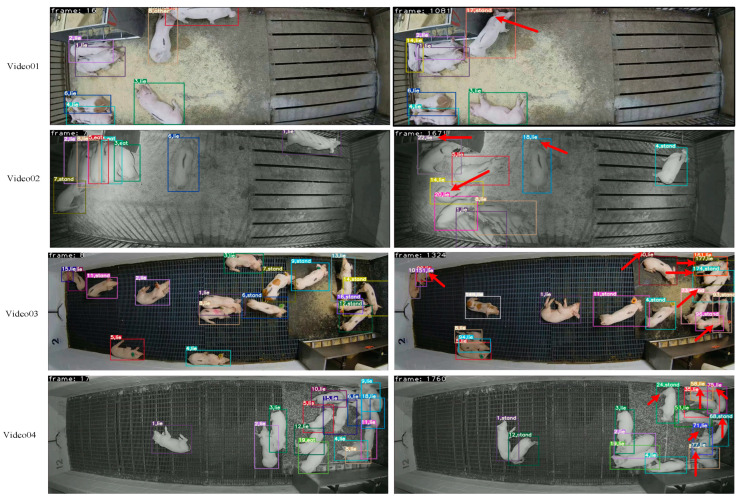
The visualized tracking results of Pig-BytetTrack in the 10 min videos. (The red arrows in the figure indicate pigs with id transformations).

**Figure 8 animals-14-03299-f008:**
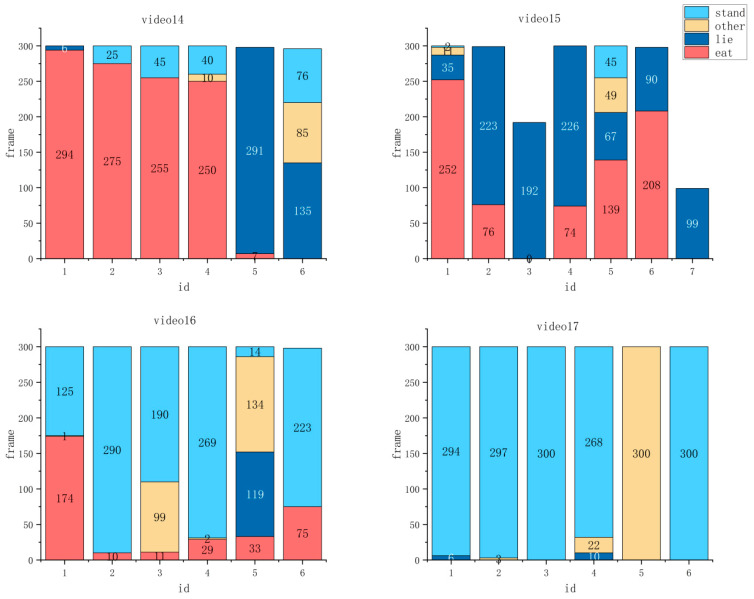
Pig behavior statistics for videos 14–17.

**Table 1 animals-14-03299-t001:** Test dataset.

Dataset	Sequence	Day	Night	Activity Level	Number of Pigs
1 min videos	10	√	–	Medium	11
	11	√	–	High	10
	12	–	√	Low	11
	13	√	–	High	11
	14	√	–	High	6
	15	√	–	Medium	6
	16	√	–	Medium	6
	17	–	√	Low	6
	18	–	√	Low	6
10 min videos	01	√	–	Medium	7
	02	–	√	Low	8
	03	√	–	High	14
	04	–	√	Low	15

The “√” indicates that the value is true, and the “–” indicates that the value is false.

**Table 2 animals-14-03299-t002:** The performance comparison between Pig-ByteTrack and the other 2 MOT methods.

Algorithms	HOTA(%) ↑	MOTA(%) ↑	IDF1(%) ↑	IDs ↓
TransTrack	49.5	87.3	68	255
ByteTrack	71.4	90.6	87.9	55
Pig-ByteTrack	**72.9**	**91.7**	**89.0**	**41**

The “↑” indicates that a higher value is better, while “↓” indicates that a lower value is preferable. and bolded results represent the method used in this study.

**Table 3 animals-14-03299-t003:** The results of each 1 min video tracking for Pig-ByteTrack.

Video	HOTA(%) ↑	MOTA(%) ↑	IDF1(%) ↑	IDs ↓
10	80.3	94.5	95.0	4
11	72.5	94.9	87.8	3
12	63.0	88.7	84.9	10
13	66.2	97.9	84.9	6
14	77.1	97.9	95.0	0
15	81.7	97.1	90.9	1
16	82.5	97.6	98.8	0
17	79.6	97.9	99.0	0
18	56.0	67.7	64.2	17
Average	72.9	91.7	89.0	41

The “↑” indicates that a higher value is better, while “↓” indicates that a lower value is preferable.

**Table 4 animals-14-03299-t004:** The 10 min video tracking results for Pig-ByteTrack.

Video	HOTA(%) ↑	MOTA(%) ↑	IDF1(%) ↑	IDs ↓
01	66.1	90.8	47.8	14
02	69.1	95.1	53.4	29
03	50.8	88.4	50.8	72
04	59.0	87.5	67.0	83
Average	59.3	89.6	53.0	198

The “↑” indicates that a higher value is better, while “↓” indicates that a lower value is preferable.

## Data Availability

All relevant data are included in the article.

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
