# Peer review of "Behavior Tracking and Analyses of Group-Housed Pigs Based on Improved ByteTrack"

_animals, 2024, doi:10.3390/ani14223299_

Round 1

Reviewer 1 Report

Comments and Suggestions for Authors

In the final part of the Introduction, the Authors wrote what they planned to do and what they did as part of the research study. In my opinion, much of this information could be formulated as the purpose of the study/research study. As a result, in the further part of the article, you can write whether the purpose/objectives were achieved. If the purpose/objectives of the study are formulated, then later (in the further part of the article) you can write whether the purpose was achieved and what research tools were needed for this. Alternatively, you can write that achieving the research purpose will require the selection of appropriate research tools or the development (improvement) of research methods used so far.

It would be worth writing what the scientific (cognitive) and practical (utilitarian) goals were that were planned to be achieved as a result of the conducted study.

In addition, I propose that based on the literature review (review of the state of knowledge), the research problem be formulated. The Introduction has already provided some issues summarizing the review of the state of knowledge, which makes it easier to identify the research problem and present this problem in one or more sentences. The research problem is usually associated with the possibility of indicating a gap in the current state of knowledge; therefore it is also worth writing what, in the Authors' opinion, may be such a gap, which the Authors want to fill with their own study/studies.

In the text, the Authors use the acronym MOT. When this acronym is used for the first time in the full text, it is worth giving its full name. The first time the full name of the acronym MOT can be found in the Abstract. In my opinion, also in the full text - starting with the Introduction - the full name of the acronym (MOT) should be written when it is used for the first time. This remark also applies to other acronyms, e.g. SORT, DeepSORT. In the case of these last two acronyms, the acronyms are given in the citation text ([2] and [3]) and probably in these cited materials the full name of the acronyms SORT and DeepSORT can be found; however, it would be very difficult for the reader to search for the full names of these acronyms in other articles. I suggest that the full names of the remaining acronyms (JDE, FairMOT, YOLOv5s, YOLO-BYTE, YOLOX-X, and others) be added to the article when first used.

The reviewed article does not have line numbers, so it is difficult to clearly indicate places in the article that I may have comments on in the review. I found line numbers only on page 8.

Analyzing the data in Table 2, I would like to ask what was the criterion for assigning the activities of "Low", "Medium" and "High" to the observed animals (pigs) or groups of animals? In other words: What was the boundary for, for example, "Medium" and "High"? Was this boundary the number of movements performed in a unit of time or the type of specific movements in a unit of time or another factor taken into account in assessing the behavior of the pigs? At the end of page 9, the Authors include the times of day: day and night, providing time ranges. In one place, the Authors write: "During the day (12:30-17:00) or night (17:00-20:00)", and in another place: "During the day (07:00-10:00) or night (20:00-07:00)". In my opinion, such time ranges introduce some confusion, if the day includes different time intervals and the night includes different time intervals. Maybe the time periods under consideration should be named differently to standardize the time issues considered in the analysis?

In the article, I could not find more detailed information about the animals (pigs) included in the study. If animals (pigs) are included in the experiment, why was there no detailed description of these animals? After all, pigs are the most important, because without them it would be impossible to conduct observations and then analyze the data. What breed were the pigs included in the study, what was the average weight of the pigs and possibly their age? In addition, it would be worth writing whether the animals included in the experiment were kept in conditions that ensured an appropriate level of welfare. What type of bedding material did the pigs lie on? What kind of floor did the animals walk on (was it a slatted floor)? What was the feeding system in the pig pen? The authors wrote about the pigs' activities, including eating, walking and lying down. That is why it is so important – in my opinion – to provide details regarding the technical equipment of pig pens that were tested in terms of developing a new research tool for observing and analyzing images in which animals (pigs) play a key role.

I would like to ask what specific pig behaviors indicate their health problems? The authors wrote about the use of the results of the conducted research to assess the health status of animals in the Simple Summary and at the end of the Conclusions. In my opinion, it would be important to explain in detail in the article how to interpret specific pig behaviors and what specific animal diseases these behaviors indicate.

Why aren't all the names of the authors of the cited articles listed in References? In the reviewed article, the authors list the first author's name with the abbreviation "et al." in References. It seemed that the rule for describing cited publications in References is different. 

Reviewer 2 Report

Comments and Suggestions for Authors

The presented article is focused on the research and development of a new method for the automatic monitoring and analysis of the behavior of pigs in a complex environment. The article is based on a systematically developed methodology and follows on from previous research in the given field of technology. I have several comments and questions about the article.

1. The Abstract should be edited to comply with the guidelines for authors, which should be: "A single paragraph of about 200 words maximum". The submitted article has an abstract of 280 words.

2. Try to reevaluate the appropriateness of some expressions in Key words: e.g. behavioral analysis algorithm; MOT; 10-minute video tracking.

3. Specify what the aim of the research is, whether it is the observation of the behavior and analysis of pigs kept in a group or the research of a method that allows this observation. Perhaps the authors could adjust the description of their research in the abstract, as they talk about Experiments here, which can create confusion as to whether the authors really measured in real pig house conditions.

4. In the first part of the Introduction, the authors state the current state of the issue and the goal, why they are dealing with this issue and what they want to achieve.

5. In the second part of article 2. Method, the authors described in detail the system of measurement, statistical evaluation and the method of processing the results, including the necessary algorithms.

6. In part 3. Experiment, called 3.1. Dataset, the authors state that "For a comprehensive analysis of group-housed pig behavior, this study collected two sets of pig behavior video datasets from different scenarios, categorized as public and private datasets.

7. I would be interested in the sensitivity and accuracy of the obtained data in terms of the size of the measured objects (pigs). Can it be used with the same accuracy for all age and weight categories of pigs?

8. It follows from the indicated research procedure that it is quite successful to identify changes in the behavior of pigs, their movement and other activities. However, in order to monitor the health and welfare of pig herds and for possible correction of problems, it would be useful to find out with high precision also the causes that lead to changes in reared animals. Did the authors manage to check what was the cause of the unusual movements?

9. One of the reasons for the emergence of problematic situations in pig farms is usually caused by thermal discomfort, e.g. excessive coldness of piglets, or, conversely, heat stress of large pigs in the summer. Given that the data used for this research is taken over, I assume that the authors did not have the opportunity to measure and detect disturbing influences acting in real stables under real conditions. Or did the authors encounter unusual behavior in pigs caused by thermal discomfort during their research?

10. I wonder if the authors have also verified their method for other types of farm animals, such as cattle or poultry.

Round 2

Reviewer 2 Report

Comments and Suggestions for Authors

The article was partially edited according to the reviewer's instructions and the authors' possibilities, and questions were answered. I recommend it for publication.